# OVI: TWIN BACKBONE CROSS-MODAL FUSION FOR AUDIO-VIDEO GENERATION

## ABSTRACT

Audio–video (AV) generation has often relied on complex multi-stage architectures or sequential synthesis of sound and visuals. We introduce OVI, a unified paradigm for audio–video generation that models the two modalities as a single generative process. By using blockwise cross-modal fusion of twin-DiT modules, OVI achieves natural synchronization and removes the need for separate pipelines or post hoc alignment. To facilitate fine-grained multimodal fusion modeling, we initialize an audio tower with an architecture identical to that of a strong pretrained video model. Trained from scratch on hundreds of thousands of hours of raw audio, the audio tower learns to generate realistic sound effects, as well as speech that conveys rich speaker identity and emotion. Fusion is obtained by jointly training the identical video and audio towers via blockwise exchange of timing (via scaled-RoPE embeddings) and semantics (through bidirectional cross-attention) on a vast video corpus. Our model enables cinematic storytelling with natural speech and accurate, context-matched sound effects, producing movie-grade video clips.

## 1 INTRODUCTION

Recent progress in video generation has come from systems—such as text-to-video (T2V), audio-to-video (A2V), and video-to-audio (V2A)—that handle one modality at a time, instead of learning audio and visuals together. In practice, however, cinematic content demands audio and video be composed jointly: speech must lip-sync and background music should match scene dynamics. Existing open-source solutions typically fix one modality and synthesize the other, relying on post hoc alignment or narrow audio-driven cases like talking-head animation. To our knowledge, truly unified one-pass audio–video generation at scale remains largely unexplored in the open literature; the only widely cited system (Google's Veo3) is closed-source and methodologically opaque.

We propose OVI, a unified generator that produces audio and video in a single pass. OVI couples two architecturally matched latent diffusion transformers (DiTs)—one for video and one for audio—via blockwise, bidirectional cross-modal attention inserted in every transformer block. A single frozen T5 encoder conditions both branches using a combined natural-language prompt, while aligned RoPE scaling reconciles their different temporal resolutions. Training proceeds in two stages: (i) initialize an audio tower mirroring the architecture of a pretrained video model and train it from scratch on large-scale, richly captioned audio to master speech and diverse sound effects; (ii) fine-tune the twin audio and video backbones with newly initialized cross-modal layers (and original attention modules) on paired audio–video data to learn synchronization without sacrificing unimodal fidelity.

**Contributions.** Guided by this framework, we make four contributions: (1) the first open-source large-scale Joint Audio Video Generation (JAVG) model capable of generating high-quality speech and non-speech audio with synchronized video; (2) a large-scale AV data pipeline (millions of videos) with strict synchronization filtering and rich captions, enabling a combined-prompt conditioning scheme (single T5 pass) that unifies semantic control across modalities; (3) an 11B symmetric twin backbone architecture that only requires simple, efficient cross-attention between modalities and scaled 3D RoPE embeddings to achieve precise cross modal temporal coupling; and (4) a scalable training recipe—audio pretraining, audio post-training, and fusion fine-tuning—that yields high-quality, synchronized clips without heuristics such as face masks or post hoc alignment.

## 2 RELATED WORK

While joint AV generation is still a relatively new field in the open-source community, various subproblems have been explored in great depth. Our review focuses on three of these subtasks: T2V generation, A2V generation, and V2A generation; we furthermore detail the efforts that have been made in joint AV generation. The unifier between the vast majority of recent models in this space is their use of the Diffusion Transformer (DiT) architecture (Peebles & Xie, 2023) inside latent space with a Flow Matching (Lipman et al., 2022) loss, along with other widely-adopted practices in generative modeling such as RoPE positional embeddings (Su et al., 2024) and Classifier-Free Diffusion Guidance (Ho & Salimans, 2022).

### 2.1 TEXT-TO-VIDEO (T2V) GENERATION

The T2V (or TI2V) task aims to generate silent video from a text prompt, optionally given a reference image (often fixed to be the first frame). The first major catalyst in this field is OpenAI's Sora (Brooks et al., 2024), which incorporates latent-space spacetime patching into a generalist diffusion transformer that handles both images and videos. The close-sourced model has led to public implementations of T2V and TI2V pipelines. One of the most impactful efforts is Wan et al. (2025), which introduces a series of open-source models that perform latent-space spatio-temporal attention and cross-attention with T5-embedded text given a fixed first-frame anchor image. The model pre-trains a 3D VAE to achieve compression at 16x16x4, which, together with their 5B Wan2.2 model, can generate realistic 720p videos at 24 fps. The natural extension to these models is to incorporate audio into the generation process, which is addressed in future A2V models.

### 2.2 AUDIO-TO-VIDEO (A2V) GENERATION

Perhaps the most common method for video generation is to condition a DiT on fixed, pre-generated audio. Tencent's HunyuanVideo (Kong et al., 2024) improves alignment with a robust data filter and an MLLM in place of T5, while HunyuanVideo-Avatar (Chen et al., 2025b) augments character and emotion by prepending a reference image, injecting emotion via cross-attention, and restricting Whisper-encoded audio features to facial regions with a mask. Tackling long-form talking-heads, Yi et al. (2025) employs Wav2Vec audio features with 3D full-attention over video, text, and reference image tokens. ByteDance's HuMo (Chen et al., 2025a) fine-tunes Wan 2.1 in two stages, first freezing most weights while appending image latents, then adding Whisper-audio cross-attention and a mask predictor. Recent models emphasize real-time streaming, e.g., Low & Wang (2025), which distills a bidirectional I2V teacher into a sparse, causal autoregressive A2V student. Unlike these pipelines, Ovi jointly generates both modalities without heuristics such as face masks, which limit generality.

### 2.3 VIDEO-TO-AUDIO (V2A) GENERATION

An alternative approach has been to fix the video modality and generate audio by conditioning on the video and text. Such V2A pipelines predict mel-spectrogram or codec latents from compressed video features using latent DiTs, often employing frame-level cross-modal attention. An early implementation of this, Diff-Foley (Luo et al., 2023), uses the standard noise-prediction objective, encoding video and audio into a shared space with a contrastively trained AV encoder, conditioning a latent diffusion model on these features, and adding a separate alignment classifier to guide inference toward temporal synchronization. Subsequent models such as Frieren (Wang et al., 2024) replace the standard diffusion with flow matching in the audio latent space, enabling faster and more stable generation. Aiming to address the lack of support for sound effects (SFX) and background music (BGM) in prior V2A models, SVA (Chen et al., 2024a) feeds a video keyframe into an MLLM to generate SFX and BGM descriptions, sending them into an audio generation and music generation model respectively, and then uses post-processing to blend the two produced waveforms. More recently, models have attempted to blend speech and generic audio generation. DeepAudio-V1 (Zhang et al., 2025) trains a CLIP-conditioned V2A module with flow matching for ambient audio, a diffusion-transformer TTS branch on transcripts, and a Mixture-of-Fusion network that fuses text, video, instructions, and V2A-predicted energy contours to fine-tune TTS into a video-to-speech (V2S) model generating synchronized speech with ambient audio. A simplified process is found in

MMAudio (Cheng et al., 2025), which performs joint attention between text, audio, and video inside a single DiT but requires an auxiliary synchronization module.

## 2.4 Joint Audio–Video Generation

The de facto for joint audio-video generation has become Google's Veo3 (Google DeepMind, 2024), a closed-source latent diffusion model capable of generating 8s video synchronized with audio. A few open-source projects have since attempted to replicate Veo3's quality and synchronization with limited success. Wang et al. (2025) introduces UniVerse-1, which uses Wan2.1 as the video backbone and the music-generation model ACE-Step as the audio backbone (Gong et al., 2025), aligning their depths by inserting interpolated transformer blocks into the shallower model and enabling blockwise cross-attention between modalities via lightweight projection layers. The model is constrained, however, due to its reliance on a pretrained music-generation model instead of training a foundational audio model, as well as the misalignment between architectures and the consequent need for block insertion, projections, and an auxiliary semantic-alignment loss to prevent degradation when fusing with video. (Liu et al., 2025) addresses the architectural mismatch by employing the same backbone for both video and audio, but requires a learned prior estimator that injects global and fine-grained latent spatio-temporal features from the text prompt into latent video via extra cross-attention in order to achieve synchronization. Ultimately, these open-source solutions demonstrate limited synchronization ability and fail to deliver consistent, high-quality video.

## 3 Data Processing Pipeline

Training a unified audio–video generator at scale requires careful construction of a large multimodal corpus. We designed a multi-stage data processing pipeline to ensure quality, diversity, and synchronization across both modalities.

### 3.1 Data Collection

To support both high-fidelity video generation and robust text-to-speech (TTS) modeling, we curate two complementary corpora: a paired audio-video corpus for learning modality alignment, as well as an audio-only corpus for acoustic pretraining and fine-tuning. The internal audio-video corpus is composed of 12M 5-second human and nonhuman data from diverse contexts. To construct the audio-only corpus, we collect both an initial pretraining subset composed of longer waveforms and a shorter-duration fine-tuning subset. This facilitates a two-stage approach, where we first train a foundational audio model and then fine-tune it on shorter, diverse data to better match deployment conditions. The pretraining data, composed of 37M waveforms up to 12-seconds long, is predominantly human speech sourced from internal collections. These longer segments emphasize linguistic diversity, prosody, and timbral variation useful for foundational acoustic modeling. The fine-tuning data, composed of waveforms that are 5-seconds long, aims to enhance the audio model to produce audio suitable for accompanying a diverse set of video scenes. As such, we emphasize modeling sound effects, drawing 2.5M 5-second waveforms from VGGSound (Chen et al., 2020), AudioSet (Gemmeke et al., 2017), and WavCaps Mei et al. (2024). To maintain TTS abilities and better align with the downstream goal, we additionally incorporate audio tracks extracted from our internal paired audio-video.

### 3.2 Audio-Video Data Preprocessing

The data processing for audio-video data is composed of four steps: (1) splitting and filtering, (2) sync detection, (3) captioning, and (4) packing.

**Splitting and filtering.** We begin by employing scene detection to isolate 121-frame clips at 24 fps that abide by certain criteria. In particular, we ensure that clips are greater than 720x720 pixel resolution, employ the optical flow model RAFT (Teed & Deng, 2020) to filter out static videos and obtain motion scores, and utilize an aesthetic predictor (Schuhmann, 2022) to remove low-quality data. We furthermore use an internal face detection model to ensure an adequate mix of single-person videos, multi-person videos, and person-free videos so that our model can learn to generate videos across a wide variety of contexts without overfitting to a particular subtask.

**Sync Detection.** We adopt the widely-used SyncNet (Chung & Zisserman, 2016) model, which uses a ConvNet architecture to learn a joint embedding between sound and mouth images, to filter out speech videos which lack sufficient audio-video synchronization. We adapt the model to handle video data on the scale of millions and run the model to produce scalar confidence and offset values. We then only retain clips with $|\text{offset}| \leq 3 \ \wedge \ \text{confidence} > 1.5$ that also meet a minimum mean volume of $-60$ decibels. We have experimentally determined that even a small quantity of out-of-sync data can impede lip-sync abilities and chose these strict criteria to minimize the risk of misaligned data.

**Captioning.** We use an MLLM to provide a verbose video caption, describing visual events interleaved with audible speech enclosed in start-of-speech and end-of-speech tags  and <E>. At the end of the caption, we ask the MLLM to provide a rich audio description, which we enclose in <AUDCAP> and <ENDAUDCAP> tags. The MLLM is provided seven evenly spaced frames from the video as well as the entire audio track, and we conducted extensive experiments to ensure the captioning included all relevant visual and audio events while respecting chronology. For clips containing speech, we ask the audio description to emphasize speaker-related acoustic attributes such as age, gender, accent, pitch, prosody, emotion, and speaking rate. For non-speech clips, the audio description instead details the sound effects, background audio, or musical elements present.

**Packing** To prepare our data for our model, we need to convert both modalities to bytes. Before doing so, we apply two final transformations to our data: we first remove any existing margins in the video and then resize the video frames (maintaining aspect ratio) to a fixed resolution of $518400 = 720 \times 720$ pixels so that our model receives consistent video frames. Finally, we convert video into an array of bytes, extracting frames at 24 fps, and convert the audio to raw wave bytes.

### 3.3 Pure Audio Data Preprocessing

For data lacking the visual modality, the preprocessing stage is simplified. We extract audio at two distinct durations—up to 12 seconds for our pretraining data and exactly 5.04 seconds (to match the duration of 121 frames at 24 fps). We employ the same MLLM as used in our audio-video data to obtain audio transcriptions (if the record does not contain audible speech, such as a pure sound effect, this is left blank) and audio descriptions.

## 4 Method

### 4.1 Architecture Overview

OVI adopts a symmetric twin backbone design with parallel audio and video branches built on an identical diffusion transformer (DiT) architecture. The video branch is initialized from Wan2.2 5B, and an identical audio branch is trained from scratch. As such, the two backbones share the same number of transformer blocks, heads, head dimensions, and FFNs, enabling symmetry at every layer, as seen in Table 1.

Table 1: Transformer hyperparameters for the OVI twin backbone.

| Model Dim | FFN Dim | Heads | Head Dim | Blocks | | |
|---|---|---|---|---|---|---|
| | | | | Self-Attn | Text Cross-Attn | AV Cross-Attn |
| 3072 | 14336 | 24 | 128 | 30 | 30 | 30 |

Each transformer block contains paired cross-attention layers, where the audio stream attends to the video stream and the video stream reciprocally attends to the audio stream. This bidirectional mechanism allows synchronization cues to be exchanged throughout the entire network. The symmetry between the audio and video towers ensures that both modalities share the same latent dimension, eliminating the need for intermediate projection layers and avoiding unnecessary parameters or computation. Importantly, it also preserves the attention structure established during unimodal pretraining, improving training stability and efficiency. In practice, the video branch uses signals from audio to enable synchronization with speech and sound effects while the audio branch grounds

speech, sound effects, and ambience in the visual context. Figure 1 details the overall architecture and fusion design.

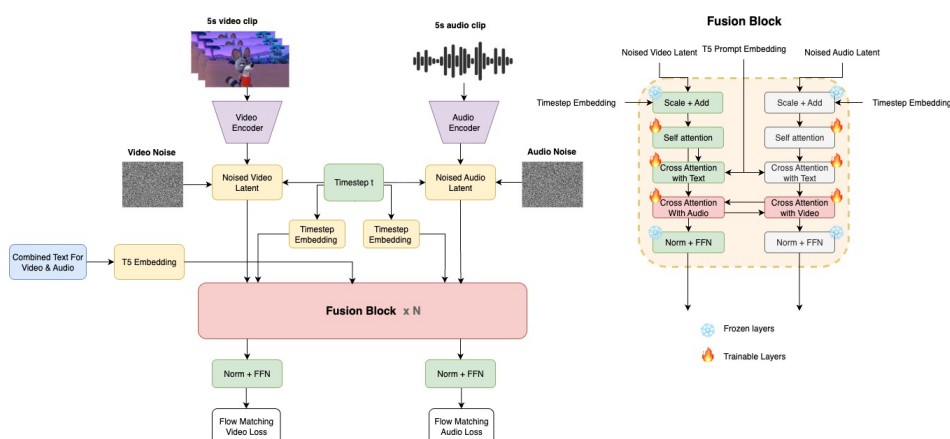

Figure 1: OVI architecture. Symmetric DiT backbones for audio and video with blockwise, bidirectional cross-attention and shared T5 conditioning from a combined prompt.

Even through the audio and video backbones share the same architecture, their temporal resolutions differ: video latents span 31 frames, while audio latents form 157 tokens ($16, \text{kHz} \times 5\text{s}/512$). To align them, we apply Rotary Positional Embeddings (RoPE) to both modalities, and, taking inspiration from Cheng et al. (2025), scale the RoPE frequencies of the audio branch by $31/157 \approx 0.197$ to match the coarser resolution of video. This scaling ensures that audio and video tokens attend to each other in a temporally consistent way. As shown in Figure 2, without scaling (left) the RoPE affinity matrix is diagonally misaligned, hindering synchronization. With scaling (right), the diagonals align sharply, providing clearer temporal correspondence.

Inspired by the scaling strategy of MMAudio (Cheng et al., 2025), OVI applies RoPE in a multi dimensional setting: MMAudio performs 1D-to-1D alignment on per-frame embeddings, whereas OVI aligns 3D spatiotemporal video tokens with 1D audio tokens, enabling token-level fusion over spatial and motion structure. Unlike MMAudio, which relies on both scaled-RoPE and Synch-Former, OVI achieves synchronization solely through bidirectional blockwise cross-attention, yielding strong lip-sync and AV coherence without external alignment modules.

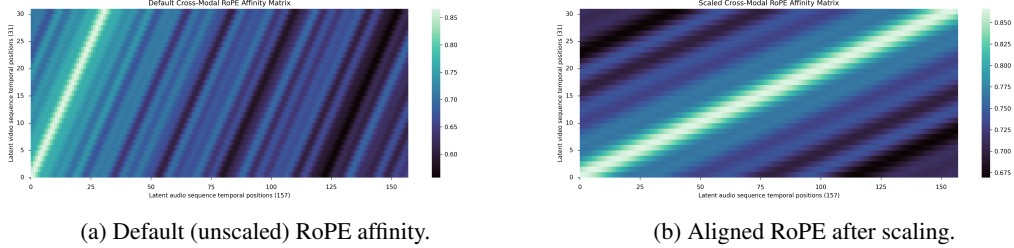

(a) Default (unscaled) RoPE affinity.          (b) Aligned RoPE after scaling.

Figure 2: Cross-modal RoPE affinity matrices before and after scaling. Scaling aligns audio and video temporal positions, improving synchronization.

OVI moreover simplifies the prompt conditioning process by utilizing a single frozen T5 encoder, applied to a combined prompt. The prompt concatenates the video caption which describes visual events interleaved with audible speech, and its T5 embedding is used independently in cross-attention with audio and video. Intuitively, details about the visual context improve the specificity and diversity of the audio, while details about the acoustic context guide facial movements and actions in the video. The single semantic context additionally simplifies training and inference and improves cross-modal alignment.

## 4.2 TRAINING STRATEGY

We train our OVI in two stages: we first initialize an audio backbone using the architecture of Wan2.2 5B and train it from scratch on speech and sound effect generation, and then we train self-attention and cross-attention layers in the joint model.

### 4.2.1 AUDIO MODEL TRAINING

For efficiency and architectural consistency with the video branch, we operate in a compact latent space using a pretrained 1D VAE from MMAudio Cheng et al. (2025). Specifically, raw audio is transformed with Short-Time Fourier Transform (STFT), converted into mel-spectrograms, and then encoded into latents by this VAE. At inference, generated latents are decoded back into spectrograms and vocoded into waveforms with BigVGAN Lee et al. (2022). We adopt only the 16kHz encoder variant, which provides an effective trade-off between efficiency and quality. We optimize a flow matching objective on audio latents: given $\mathbf{z}_1^a \sim p_{\text{data}}^a$ and $\mathbf{z}_0^a \sim \mathcal{N}(0, \mathbf{I})$, we form the linear interpolant $\mathbf{z}_t^a = (1-t)\mathbf{z}_0^a + t\mathbf{z}_1^a$ with $t \sim \mathcal{U}[0,1]$ and train a velocity predictor $\mathbf{v}_\theta^a(\mathbf{z}_t^a, t, \mathbf{c}_{\text{text}})$ toward the target $\mathbf{z}_1^a - \mathbf{z}_0^a$,

$$\mathcal{L}_{\text{FM}}^a = \mathbb{E}_{t, \mathbf{z}_1^a, \mathbf{z}_0^a} \left[ \|\mathbf{v}_\theta^a(\mathbf{z}_t^a, t, \mathbf{c}_{\text{text}}) - (\mathbf{z}_1^a - \mathbf{z}_0^a)\|_2^2 \right]. \tag{1}$$

Our audio tower OVI-AUD is trained in two substages: an initial pretraining stage of up to 12-second waveforms, and a fine-tuning stage of up to 5-second waveforms. To avoid re-adaptation when transitioning to the audio-video finetuning stage and eliminate the need to maintain multiple scales for audio RoPE, we applied scaled RoPE positional embeddings to all attention layers.

**Audio Pretraining.** The audio backbone is pretrained from scratch on hundreds of thousands of hours of primarily speech data up to 12 seconds in duration. During pretraining, we use variable-length audio to maximize coverage of diverse acoustics, providing the audio backbone with broad exposure to natural variability in duration and content. The long-form raw audio enables the model to generate consistent audio that respects speaker traits such as pitch and emotion.

**Audio Fine-tuning** We next fine-tune the pretrained audio model with padded 5.04-second waveforms to produce audio compatible with our generated video. This step ensures that the audio backbone aligns with the distribution expected in multimodal fusion training, while retaining the generalization capacity learned from large-scale diverse pretraining. At this phase, a variety of sound effects are also introduced into the training mix, enabling the OVI-AUD to serve as a foundational audio model for AV generation.

### 4.2.2 AUDIO–VIDEO MODEL TRAINING

**Fine-tuning attention layers.** We combine pretrained audio and video backbones, initializing cross-modal attention from scratch while freezing all FFNs to reduce memory, leaving 5.7B of 11B parameters trainable. By fine-tuning only unimodal self-attention and cross-attention modules (text-to-modality and modality-to-modality), we align audio and video while preserving their pretrained representations. Building on Eq. equation 1, we train on paired AV latents $(\mathbf{z}_1^v, \mathbf{z}_1^a)$ with independent noises $(\mathbf{z}_0^v, \mathbf{z}_0^a)$ and a shared $t \sim \mathcal{U}[0,1]$, defining $\mathbf{z}_t^m = (1-t)\mathbf{z}_0^m + t\mathbf{z}_1^m$ for $m \in \{v, a\}$. Each backbone predicts a velocity conditioned on text and the other modality via bidirectional cross-attention, and we apply the same FM objective per modality; the total loss is a weighted sum

$$\mathcal{L}_{\text{total}} = \lambda_v \mathcal{L}_{\text{FM}}^v + \lambda_a \mathcal{L}_{\text{FM}}^a, \quad \lambda_v = 0.85, \ \lambda_a = 0.15.$$

Paired sampling and a shared timestep encourage the model to learn audio–visual correspondences (e.g., lip-sync, action–sound alignment) without explicit sync losses. At inference, both branches share the same $t$ schedule and are jointly integrated with a single ODE solver.

## 4.3 IMPLEMENTATION DETAILS

The audio pretraining phase described in subsection 4.2.1 was conducted for 50k steps with a batch size of 2880 and a learning rate of $1 \times 10^{-4}$. We used the AdamW optimizer with parameters $\beta_1 = 0.9, \beta_2 = 0.999, \epsilon = 10^{-8}$. Upon convergence of the audio tower, denoted as OVI-AUD, we proceeded with the audio-video fusion training phase as in subsection 4.2.2. We trained the partially

frozen fusion model for 40k steps with a batch size of 768 and a learning rate of $5 \times 10^{-5}$, using AdamW optimizer with parameters $\beta_1 = 0.9, \beta_2 = 0.95, \epsilon = 10^{-8}$. All models were trained at `bf16` precision leveraging DeepSpeed (Rasley et al., 2020) for efficient sharded distributed Data Parallel (DP) training. We employ the UniPC (Zhao et al., 2023) solver as we experimentally verify that it improves stability compared to a standard Euler solver.

## 5 EXPERIMENTS

### 5.1 CROSS-MODAL ATTENTION VISUALIZATIONS.

We visualize A2V cross-attention maps by averaging token alignments and projecting them into pixel heatmaps, highlighting where audio attends in the visual scene. As shown in Figure 3, speech emphasizes mouths, drumming highlights drums, and animal sounds align with the source body parts, illustrating that the fusion model effectively synchronizes audio with relevant visual cues.

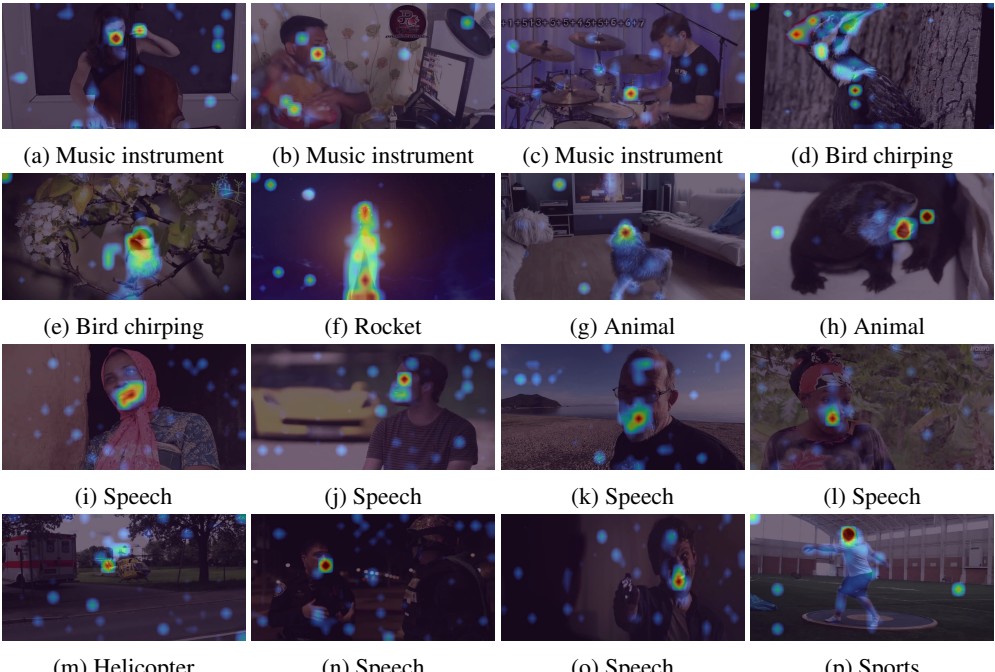

| (a) Music instrument | (b) Music instrument | (c) Music instrument | (d) Bird chirping |
| (e) Bird chirping | (f) Rocket | (g) Animal | (h) Animal |
| (i) Speech | (j) Speech | (k) Speech | (l) Speech |
| (m) Helicopter | (n) Speech | (o) Speech | (p) Sports |

Figure 3: A2V cross-attention visualizations. Heatmaps highlight pixels most attended by audio tokens. Brighter regions correspond to stronger attention.

### 5.2 COMPARED METHODS

Mirroring our two-stage training phase, we evaluate each stage independently. After the audio pre-training stage, we assess the audio generation capabilities of the audio tower (OVI-AUD) against state-of-the-art baselines in both text-to-audio (T2A) and text-to-speech (TTS).For text-to-audio (T2A), baselines include GenAU (Haji-Ali et al., 2024), TANGO 2 (Majumder et al., 2024), Make-An-Audio 2 (Huang et al., 2023), AudioLDM2 (Liu et al., 2024), and MMAudio. For text-to-speech (TTS), we evaluate against Fish Speech (Liao et al., 2024) and F5-TTS (Chen et al., 2024b), CosyVoice (Du et al., 2024), FireRedTTS (Guo et al., 2024).

In the second stage, we evaluate the joint audio-video generation (JAVG) capabilities of OVI, comparing against JavisDiT and UniVerse-1. We also compare the video generation quality relative to the pretrained Wan2.2 video model to ensure that the JAVG ability did not come at the expense of degraded video performance. Furthermore, to quantify the advantage of a single unified JAVG model over sequential, modality-specific pipelines, we compare against two pipeline baselines: (1) image-to-video, followed by video-to-audio generation, and (2) text-to-audio, followed by audio-to-video generation. For text prompts that contain spoken content, we first synthesize speech using

Fish Speech (choosing reference voices based on gender) and then generate audio-driven video using Wan2.2 14B S2V (Gao et al., 2025a). For prompts without speech, we instead use Wan2.2 5B for image-to-video generation and subsequently apply MMAudio to synthesize the corresponding sound effects or music. For simplicity, we refer to the combination of these pipelines collectively as AV-PIPE.

### 5.3 EVALUATION METRICS

We generated videos for all 205 image-text pairs in the Verse-Bench Set 1 dataset (Wang et al., 2025), using OVI, JavisDiT, UniVerse-1, and AV-PIPE, and ran a blind study with 50 participants. Each participant evaluated 100 randomly sampled prompt-matched pairs comparing OVI to a baseline. We report Pairwise Win Rate (PWR) over audio quality, video quality, and AV synchronization.

For each pair of videos, participants were asked the following questions:

- **Audio quality**: Based on your assessment of sound clarity, naturalness, and absence of noise or distortion, which clip demonstrates higher overall audio quality?

- **Video quality**: Considering factors such as visual clarity, sharpness, motion smoothness, and absence of visual artifacts, which clip demonstrates higher overall video quality?

- **AV sync**: Taking into account how accurately the audio aligns with visible speech movements and other relevant visual events, as well as how well the audio content semantically matches the actions, mood, or context of the video, which clip demonstrates better overall audio–video alignment?

In addition to human evaluation, we report quantitative measures of audio–video synchronization using Synchformer-C and Synchformer-O, corresponding to the top-1 confidence and temporal offset predicted by SynchFormer (Iashin et al., 2024) respectively. Following SEEDANCE 1.0 (Gao et al., 2025b), we omit FVD, which is known to be unstable and poorly correlated with human-judged video quality.

Following MMAudio's T2A protocol, we evaluate OVI-AUD with $FD_{PANNs}$ (Kong et al., 2020), $FD_{VGG}$ (Hershey et al., 2017), Inception Score (IS) (Salimans et al., 2016), and CLAP (Wu et al., 2023). $FD_{PANNs}$ and $FD_{VGG}$ compare distributional distances with different pretrained extractors, IS measures perceptual quality, and CLAP evaluates text–audio semantic alignment. For TTS, we report linguistic correctness using Word Error Rate (WER).

### 5.4 RESULTS

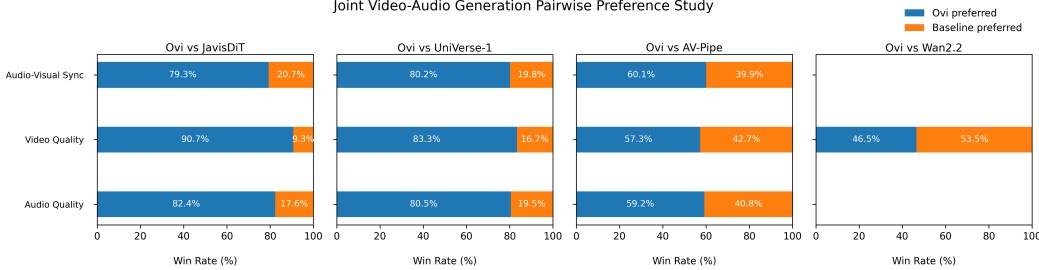

Figure 4: Pairwise win rate (PWR) results of OVI compared against baselines on Verse-Bench Set 1. Higher values indicate stronger human preference for OVI.

As shown in Figure 4, OVI achieves a clear and consistent preference over both JavisDiT and UniVerse-1 across all three evaluation dimensions: audio quality, video quality, and audio-video synchronization. Notably, the margins are substantial, with participants overwhelmingly favoring OVI. This indicates that our unified design and training framework does not simply maintain competitive performance, but pushes the boundaries of open-research joint audio-video generation, bringing the community significantly closer to the capabilities demonstrated by frontier models such

as Veo3 (Google DeepMind, 2024). We note, however, a slight degradation in video quality relative to the Wan2.2 base model, which is expected given that our joint training relies on a narrower audio-video dataset compared to the large-scale pretraining corpus used for Wan2.2. Importantly, this trade-off is marginal and does not diminish the overall superiority of OVI in joint audio-video generation.

Beyond model-to-model comparisons, OVI also outperforms the AV-PIPE baseline across all evaluation dimensions. Although the margins are smaller than those against JavisDiT and UniVerse-1, which reflects the relative strength of a well-designed cascaded pipeline, the unified approach of OVI still yields clearer audio–video coherence and higher overall perceptual quality.

Table 2: SynchFormer results on Verse-Bench Set 1. Synchformer-C is the predicted synchronization confidence (higher is better), and Synchformer-O is the temporal offset in seconds (lower is better).

| | UniVerse-1 | JavisDiT | AV-Pipe | OVI |
|---|---|---|---|---|
| Synchformer-C ↑ | 0.2732 | 0.3750 | **0.5024** | 0.4922 |
| Synchformer-O ↓ | 0.86 | 0.59 | **0.48** | **0.48** |

Table 2 shows that OVI substantially outperforms JavisDiT and UniVerse-1, achieving both higher synchronization confidence and lower temporal offset. When compared with the AV-PIPE baseline, OVI exhibits comparable overall synchronization performance. A more detailed breakdown reveals that OVI performs notably better on speech-driven prompts, whereas AV-PIPE shows an advantage on non-speech cases. This pattern is expected because non-speech samples in AV-PIPE are generated using MMAudio. Since MMAudio incorporates the frozen SynchFormer visual encoder as part of its architecture, it naturally learns to align its audio outputs with SynchFormer's visual latent space, which can lead to inflated scores when evaluated using the same model. Crucially, the Pairwise Win Rate (PWR) results demonstrate that, independent of SynchFormer's inherent bias, human evaluators consistently prefer OVI, confirming its clear superiority in joint audio–video generation.

Table 3: Results of audio evaluation. All T2A metrics follow the evaluation protocol of MMAudio, and baseline results are directly copied from that work. WER was computed on Seed-TTS test-en dataset(Anastassiou et al., 2024), and baseline results are directly copied from Chen et al. (2024b)

| Type | Model | T2A Metrics | | | | TTS Metric |
|---|---|---|---|---|---|---|
| | | $FD_{PANNs}$ ↓ | $FD_{VGG}$ ↓ | IS ↑ | CLAP ↑ | WER ↓ |
| T2A | GenAU-Large | 16.51 | **1.21** | 11.75 | 0.285 | - |
| | TANGO 2 | 19.77 | 2.74 | 8.45 | 0.264 | - |
| | Make-An-Audio 2 | 15.34 | 1.27 | 9.58 | 0.251 | - |
| | AudioLDM 2-L | 32.50 | 5.11 | 8.54 | 0.212 | - |
| | MMAudio-L | **15.04** | 4.03 | **12.08** | **0.348** | - |
| TTS | Fish Speech | - | - | - | - | **0.008** |
| | F5-TTS | - | - | - | - | 0.018 |
| | CosyVoice | - | - | - | - | 0.034 |
| | FireRedTTS | - | - | - | - | 0.038 |
| Unified | OVI-AUD (ours) | 18.03 | 5.02 | 11.20 | 0.224 | 0.035 |

As shown in Table 3, our unified audio model, OVI-AUD, capable of both T2A and TTS, achieves performance comparable to dedicated state-of-the-art models on their respective metrics. While it is expected that a unified model may not surpass specialized models optimized for a single task, attaining competitive results across both domains demonstrates that OVI-AUD is sufficiently strong for its primary role as a foundation for audio-video fusion. Crucially, unified audio generation is particularly important for joint audio-video modeling, since real-world videos often contain both complex sound effects and coherent speech capabilities that specialized models are unable to support.

## 5.5 Ablation Study

The initial design of our audio tower (OVI-AUD) incorporated both a CLAP text encoder and a T5 text encoder. The motivation was to disentangle T2A and TTS tasks by providing separate text embeddings, thereby preventing the two objectives from interfering with each other adversely. In practice, however, we observed that this separate embedding setup constrained the model's ability to generate cohesive outputs: while it could handle either sound effects or speech in isolation, it struggled to integrate them into a unified and coherent audio stream.

To address this, we adopted the combined text prompt approach described in subsection 4.1, where both speech transcripts and textual audio descriptions are fused into a single cohesive T5 text embedding. This modification preserved the linguistic correctness of the model as seen from the comparable WER, while significantly improving the audio fidelity and alignment metrics (FD, IS and CLAP), as seen in Table 4. More importantly, the unified text embedding also streamlined joint audio-video generation, as both the audio and video towers could now condition on the same T5 text representation, simplifying cross-modal modeling and strengthening multimodal coherence.

Table 4: Ablation study of audio tower design, specifically using a separate CLAP encoder for non-speech audio descriptions

| Variant | $FD_{PANNs} \downarrow$ | $FD_{VGG} \downarrow$ | IS $\uparrow$ | CLAP $\uparrow$ | WER $\downarrow$ |
|---|---|---|---|---|---|
| OVI with CLAP | 20.78 | 7.13 | 8.34 | 0.190 | **0.033** |
| OVI | **18.03** | **5.02** | **11.20** | **0.224** | 0.035 |

## 6 Limitations and Conclusion

**Limitations.** In its current form, OVI is tuned to short (5s) 720p/24 fps clips, which leaves minute-scale narratives, inter-shot transitions, and global story consistency out of scope. Future work could explore methods to increase duration; for example, stitching multiple 5s "chunks" together by training a chunk-wise causal audio model and pairing it with a causal video backbone that conditions on the last frame of the previous chunk. The symmetric 5B-per-branch design with dense, blockwise fusion also requires significant time per sampling step, which is exacerbated by the additional forward pass for each step due to CFG. Distillation via a framework such as DMD2 (Yin et al., 2024) could reduce the effective number of sample steps needed. On the audio side, the 16 kHz path through a fixed 1D-VAE constrains bandwidth and spatial realism, so high-fidelity music, spatial cues, and subtle timbre can be flattened. By replacing the fixed 1D-VAE with a higher-bandwidth latent or performing bandwidth extension in post-processing, audio quality could be further improved.

**Conclusion.** We introduced OVI, a framework for unified audio–video generation that treats the two modalities as a single generative object. Architectural symmetry and blockwise bidirectional fusion allow timing and semantics to be learned jointly rather than via sequential pipelines, while a pretrained foundational audio tower—capable of both speech and diverse sound effects—supports general synchronization without heuristic add-ons (e.g., face masks or auxiliary sync modules). Empirically, this unified formulation is competitive and produces coherent, synchronized outputs, establishing a practical template for simple and scalable AV generation. Ultimately, our twin backbone architecture proves effective and sets a direction for future AV systems.

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
