# OpenReview forum: "Ovi: Twin Backbone Cross-Modal Fusion for Audio-Video Generation"
_ICLR.cc/2026/Conference — Submitted to ICLR 2026_

### Official Review · Reviewer_pr5F · 2025-10-24

**Soundness:** 2
**Presentation:** 3
**Contribution:** 3
**Rating:** 4
**Confidence:** 3

**Summary:**

This paper introduces OVI, a unified audio-video generation model using symmetric twin DiT backbones. The model employs blockwise cross-modal fusion and scaled-RoPE for temporal alignment. It is trained in stages: first pretraining a foundational audio tower, then jointly finetuning the twin backbones on a large-scale audio-video corpus.

**Strengths:**

1. The work presents a one-stage, end-to-end framework for joint audio-video generation. This approach successfully enables lip-synchronized speech generation.
2. The authors have developed a comprehensive data processing pipeline for large-scale audio-video data. This pipeline notably includes strict synchronization filtering using SyncNet and a unified captioning strategy.
3. The paper provides cross-modal attention visualizations (Figure 3). These qualitatively demonstrate that the model learns meaningful alignments between modalities.

**Weaknesses:**

1. The supplementary material lacks comparative results (e.g., generated videos) against other methods.
2. The subjective evaluation is missing an assessment of semantic alignment.
3. The objective evaluation lacks metrics for video quality, text-video alignment, and audio-video alignment/synchronization.
4. Comparisons against several other relevant methods are missing [3] [4].
5. The paper's objective evaluation is incomplete. It misses an objective comparison table for the joint audio-video generation task against baselines [1][2][3][4].

[1] JavisDiT: Joint Audio-Video Diffusion Transformer with Hierarchical Spatio-Temporal Prior Synchronization

[2] UniVerse-1: Unified Audio-Video Generation via Stitching of Experts

[3] A simple but strong baseline for sounding video generation: Effective adaptation of audio and video diffusion models for joint generation.

[4] MM-Diffusion: Learning Multi-Modal Diffusion Models for Joint Audio and Video Generation

**Questions:**

1. Will the training dataset be made publicly available?
2. What would happen if the video and audio branches used their own separate prompts?
3. The paper fails to provide ablation studies for its most innovative and critical components, such as the effect of the scaled-RoPE technique.

---

> ### Author Response · Authors · 2025-11-23
> **Rebuttal - Part I**
>
> We thank the reviewer for highlighting the end-to-end design, lip synchronization, and data pipeline contributions. We have revised our paper to address their critiques and hope the following clarifications will address the reviewer’s concerns.
>
> **1. Comparative results not provided in supplementary materials**
>
> > The supplementary material lacks comparative results (e.g., generated videos) against other methods.
>
> To address (1), we create two anonymous websites: the first, which can be found at https://anonymous-iclr-2026.github.io/ovi/, presents matching videos generated by Ovi, Javis and UniVerse-1, these 25 samples are uniformly randomly selected to fit upload constraints and ease reviewer inspection, which can be found as zips in the supplementary materials as well. Full 205-prompt results are available upon request; the second, which can be found at https://anonymous-iclr-2026.github.io/sequential-comparison/, compares Ovi and sequential pipelines (AV-Pipe).
>
> **2. Limited Baseline Comparisons and Evaluation Gaps**
>
> > - The subjective evaluation is missing an assessment of semantic alignment.
> > - The objective evaluation lacks metrics for video quality, text-video alignment, and audio-video alignment/synchronization.
> > - The paper’s objective evaluation is incomplete. It misses an objective comparison table for the joint audio-video generation task against baselines.
>
> We thank the reviewer for these constructive comments and for pointing out the gaps in our evaluation. Regarding point (1), we note that another reviewer encouraged us to provide additional details about our Pairwise Win Rate survey. In the revised manuscript, we have clarified that one of our key survey questions directly targets semantic and contextual alignment. Specifically, participants were asked:
>
> *"Taking into account how accurately the audio aligns with visible speech movements and other relevant visual events, as well as how well the audio content semantically matches the actions, mood, or context of the video, which clip demonstrates better overall audio–video alignment?"*
>
> This question explicitly covers semantic alignment between audio and video, and therefore we believe our subjective evaluation already captures this aspect to a meaningful extent.
>
> We also appreciate the reviewer’s suggestions regarding objective evaluation metrics. In the revised paper, we have incorporated a table on objective evaluation for the joint audio–video generation (JAVG) task. Following Seedance 1.0, we omit FVD, which is known to be unstable and poorly correlated with human-judged video quality, and instead rely on comprehensive pairwise evaluations to evaluate video generation quality. We now include, in Sec 5.4 of our paper, Synchformer results on Verse-Bench Set 1, where Synchformer-C is the predicted synchronization confidence (higher is better), and Synchformer-O is the temporal offset in seconds (lower is better). These metrics provide objective measurements of audio–video alignment and complement our subjective study.
>
> We hope these additions address the reviewer’s concerns and strengthen the completeness of our evaluation.
>
> |  | UniVerse-1 | JavisDiT | AV-Pipe | Ovi|
> |---|---:|---:|---:|---:|
> | Synchformer-C ↑ | 0.2732 | 0.3750 | **0.5024** | 0.4922 |
> | Synchformer-O ↓ | 0.86 | 0.59 | **0.48** | **0.48** |

---

> > ### Author Response · Authors · 2025-11-23
> > **Rebuttal - Part II**
> >
> > **3. Dataset Release**
> >
> > > Will the training dataset be made publicly available?
> >
> > We thank the reviewer for their question. We will release model checkpoints and inference code, and plan to further release our data processing and filtering pipeline, as well as training scripts. Portions of our dataset may be made available subject to licensing review. Our firm goal is to provide the most complete open-source AV generation stack to date.
> >
> > **4. Incomplete Ablation Studies**
> >
> > > 1. What would happen if the video and audio branches used their own separate prompts?
> > > 2. The paper fails to provide ablation studies for its most innovative and critical components, such as the effect of the scaled-RoPE technique.
> >
> > For point (1), we clarify that the benefit of unified prompting is demonstrated through an ablation already included in the paper in Table 4. Our initial design used separate text encoders—T5 for speech and CLAP for non-speech audio—to disentangle TTS and T2A signals. However, this dual-conditioning setup fragmented semantics across modalities and led to weaker audio generation quality (higher FD, lower IS/CLAP), despite slightly lower WER. These findings motivated the single-embedding formulation used in Ovi.
> >
> > We adopt a single unified prompt because it provides both modalities with consistent and complete semantic information, avoids fragmenting the conditioning signal, and aligns with standard practice in unified multimodal generation. It is also architecturally simpler and naturally supports coherent cross-modal alignment.
> >
> > For point (2), concerning a potential ablation on scaling RoPE embeddings: the video and audio streams exhibit inherent temporal alignment, and thus applying consistent scaled RoPE across modalities during cross-attention is the only configuration that preserves temporal coherence. Using unscaled RoPE would introduce misalignment by construction. Figure 2 in our main submission, which compares affinity matrices under scaled and unscaled RoPE, further supports this rationale. Consequently, we did not view this as a meaningful ablation to include.

---

### Official Review · Reviewer_XP5Y · 2025-10-29

**Soundness:** 1
**Presentation:** 2
**Contribution:** 1
**Rating:** 2
**Confidence:** 4

**Summary:**

This paper presents a joint audio-video generation model.
Starting from a pre-trained video generation model (Wan2.2), the audio backbone is designed with an identical architecture, which simplifies the design of the cross-modal interaction modules.
The model is trained in two stages: the audio backbone is first trained from scratch on speech and sound-effect datasets, followed by joint training of the audio and video branches on audiovisual datasets.
Subjective evaluations demonstrate that the proposed model outperforms existing open-source joint audio-video generation models.

**Strengths:**

- The proposed model achieves state-of-the-art quality compared to existing open-source joint audio-video generation models.
- The architecture is simple yet effective, yielding improved generation performance.
- A new audiovisual dataset construction pipeline is introduced, producing well-synchronized audio-video pairs with rich captions, which could serve as a valuable contribution to the community.

**Weaknesses:**

**Lack of methodological novelty**.
The proposed approach replicates prior work (especially JavisDiT), raising concerns about the method's originality. Specific overlaps include:

- The overall modeling framework and two-stage training strategy closely follow JavisDiT. JavisDiT also employs an identical architecture for the audio backbone with video branch (while it is based on OpenSora rather than Wan2.2) and uses a similar two-step training procedure: audio pre-training followed by joint audiovisual training.
- The paper explains the limitation of JavisDiT as requiring "learned prior estimator ... in order to achieve synchronization" (Sec 2.4), but it is unclear why this is problematic or how severe this limitation actually is.
- The proposed unified prompt conditioning mechanism differs from JavisDiT, but there is no direct comparison demonstrating its benefit.
- The proposed RoPE configuration appears to be identical to that of MMAudio (see Fig. A5 in the MMAudio paper), which does not solely establish the novelty.

Given these similarities and lack of justification of the proposed design choices, the contributions (2), (3), and (4) in the introduction are not convincingly supported.

**Insufficient experimental evidence**.
The experiments are limited and do not clearly demonstrate the advantages of the proposed model.

- Details of the subjective evaluation are missing. How many samples were evaluated per participant? Which prompts (or Verse-Bench splits) were used for generation? What questions were employed for each evaluation criterion?
- Objective evaluation is limited to T2A and TTS tasks and focuses only on audio quality. Since the proposed method jointly generates both audio and video, it would be better to include evaluation of video quality (e.g., FVD[1], CLIP score[2], or VBench[3]) and audiovisual alignment (e.g., ImageBind[4] similarity, AV-Align[5], or DeSync[6]).
- Fig.4 indicates that Ovi underperforms Wan2.2 for video generation, while Table 2 shows that Ovi underperforms most existing T2A or TTS models for audio generation. Based solely on these results, it is difficult to identify the strengths of the proposed approach. It would be more convincing to compare Ovi with a sequential baseline (e.g., Wan2.2 for T2V followed by MMAudio-L for V2A, or FishSpeech for TTS) to demonstrate the benefit of joint modeling.

[1] "FVD: A new metric for video generation," ICLRW, 2019
[2] "Learning transferable visual models from natural language supervision," ICML, 2021
[3] "VBench: Comprehensive Benchmark Suite for Video Generative Models," CVPR 2024
[4] "Imagebind: One embedding space to bind them all," CVPR, 2023
[5] "Diverse and aligned audio-to-video generation via text-to-video model adaptation," AAAI, 2024
[6] "Synchformer: Efficient synchronization from sparse cues," ICASSP, 2024

**Questions:**

See weaknesses above, particularly regarding the novelty and experimental setup.

---

> ### Author Response · Authors · 2025-11-23
> **Rebuttal - Part I**
>
> We thank the reviewer for their recognition of our contributions and for their thoughtful, constructive feedback. We have worked hard to thoroughly address all of the reviewer’s concerns in the body of our paper and present the following clarifications to expand on the novelty and evidence behind our method.
>
> **1. Methodological Novelty**
>
> We appreciate the reviewer’s concern about novelty and would like to go through each of their points to show that, beyond surpassing other open-source models in generation quality, Ovi also exhibits significant methodological uniqueness.
>
> > - The overall modeling framework and two-stage training strategy closely follow JavisDiT. JavisDiT also employs an identical architecture for the audio backbone with video branch… and uses a similar two-step training procedure.
> > - The paper explains the limitation of JavisDiT as requiring “learned prior estimator ... in order to achieve synchronization” (Sec 2.4), but it is unclear why this is problematic or how severe this limitation actually is.
>
> The reviewer is right to point out certain similarities between the architecture and training methods of Ovi and JavisDiT. There are, however, differences that, when taken together, allow Ovi to produce more consistent, synchronized, and expressive video and audio. An updated set of our contributions can be found at the end of our paper’s Introduction (Sec 1).
>
> The biggest difference comes from the actual conditioning of the model. In addition to cross-attention with T5 embeddings, JavisDiT requires an additional cross-attention module with spatio-temporal priors, which are obtained by feeding the embeddings of a second encoder (ImageBind) into a prior estimator that must be explicitly trained via contrastive learning. In addition to introducing additional complexity and requiring a separate round of training, the priors used in cross-attention introduce a misalignment risk: they are *external* to the generative model, and because they do not involve audio/video latents or follow the diffusion path during inference, they can contradict the model’s learned structure. Moreover, these priors are *sampled* from a distribution \(z_{\mathrm{prior}} \sim \mathcal{N}(\mu(\mathrm{text}), \sigma(\mathrm{text}))\), which means that every video gets different sampled priors even for the same exact prompt. There is furthermore no guarantee that the priors match the semantics from the T5 embeddings, as they do not live in the same representation space. Finally, because the priors are learned separately via contrastive learning, the model’s representation power is limited by the accuracy of the priors, which may not scale up to complex scenarios.
>
> In contrast, as Ovi’s alignment is learned organically through cross-modal attentions rather than constrained by handcrafted hierarchical priors, the model scales much better with data volume and exhibits stronger emergent multimodal consistency. As demonstrated in our PWR results, Ovi achieves substantially improved speech generation and lip synchronization, higher perceptual preference in both audio and video quality, and strong motion range and cinematography fidelity. These results suggest that Ovi’s synchronization mechanism is not only architecturally different, but also a more scalable formulation that benefits from larger end-to-end AV training corpora.
>
> Finally, Ovi is a foundational video and audio generation model: the audio branch is pretrained on 37M waveforms and fine-tuned on 2.5M diverse sound effects as well as audio from 12M video data points, used in fusion training. Meanwhile, JavisDiT is trained on 0.8M audio samples and 0.6M video data points. Importantly, neither MMAudio nor JavisDiT can generate human speech at all, while Ovi is the first open-sourced model that is able to generate high-quality non-speech and speech audio, as well as multi-turn dialogues with accurate synchronization and lipsyncing capabilities. The clear difference in generation quality can be observed in our expanded evaluations – found at https://anonymous-iclr-2026.github.io/ovi/. Ovi is therefore the only proven large-scale joint audio-video generation model available to the open-source community.

---

> > ### Author Response · Authors · 2025-11-23
> > **Rebuttal - Part II**
> >
> > > The proposed unified prompt conditioning mechanism differs from JavisDiT, but there is no direct comparison demonstrating its benefit.
> >
> > We clarify that the benefit of unified prompting is demonstrated through an ablation already included in the paper in Table 4. Our initial design used separate text encoders—T5 for speech and CLAP for non-speech audio—to disentangle TTS and T2A signals. However, this dual-conditioning setup fragmented semantics across modalities and led to weaker audio generation quality (higher FD, lower IS/CLAP), despite slightly lower WER. These findings motivated the single-embedding formulation used in Ovi.
> >
> > We adopt a single unified prompt because it provides both modalities with consistent and complete semantic information, avoids fragmenting the conditioning signal, and aligns with standard practice in unified multimodal generation. It is also architecturally simpler and naturally supports coherent cross-modal alignment.
> >
> > > The proposed RoPE configuration appears to be identical to that of MMAudio (see Fig. A5 in the MMAudio paper), which does not solely establish the novelty.
> >
> > We thank the reviewer for their observation and would like to emphasize some critical differences between the RoPE configuration of MMAudio and Ovi. MMAudio encodes video as a *single* embedding per frame and performs 1D-to-1D alignment, with both modalities represented as flat temporal sequences. In contrast, Ovi performs alignment between compressed 3D spatio-temporal video tokens (\(H \times W \times T\)) and raw 1D audio tokens. Scaled-RoPE in Ovi therefore maps across heterogeneous geometries rather than between matched 1D temporal sequences, enabling token-level cross-attention over full spatial and motion structure rather than frame-level conditioning. This difference is necessary for maintaining cinematography, motion dynamics, and lip articulation during joint generation.
> >
> > Furthermore, MMAudio requires *both* scaled-RoPE and a SynchFormer-based sync module to achieve alignment, whereas Ovi achieves synchronization *solely through bidirectional blockwise cross-attention*. No auxiliary sync model is used. Ovi’s achievement of high-quality video-audio synchronization and lipsyncing demonstrates that the simplified design is highly scalable with large training data.

---

> ### Author Response · Authors · 2025-11-23
> **Rebuttal - Part III**
>
> **2. Experimental Evidence**
>
> We appreciate the reviewer’s request for greater experimental evidence and have made numerous strides to address their concerns. The requested details are now present in the Experiments section of our paper.
>
> > Details of the subjective evaluation are missing.
>
> The requested details are now added in full in Sec 5.3 of our paper. We clarify that we generated videos for all 205 image-text pairs in the Verse-Bench Set 1 dataset, using  Ovi, JavisDiT, UniVerse-1, and AV-Pipe, and ran a blind study with 50 participants. Each participant evaluated 100 randomly sampled prompt-matched pairs comparing OVI to a baseline. We furthermore provide, verbatim, the questions posed to the participants in the PWR evaluation.
>
> > Objective evaluation is limited to T2A and TTS tasks and focuses only on audio quality. Since the proposed method jointly generates both audio and video, it would be better to include evaluation of video quality.
>
> We appreciate the reviewer’s concern for objective evaluations on video quality. We first direct them to an anonymous website, found at https://anonymous-iclr-2026.github.io/ovi/, which presents matching videos generated by Ovi, JavisDiT and UniVerse-1. The site contains the generations corresponding to 25 randomly selected evaluation prompts out of 205 in Verse-Bench Set 1. These 25 samples are uniformly randomly selected to fit upload constraints and ease reviewer inspection, which can be found as zips in the supplementary materials as well; full 205-prompt results are available upon request. The clear conclusion is that Ovi exhibits not only superior video quality but also greater expressiveness and sync.
>
> Following Seedance 1.0, we omit FVD, which is known to be unstable and poorly correlated with human-judged video quality, and instead opt to provide detailed pairwise evaluations. We furthermore now include, in Sec 5.4 of our paper, Synchformer results on Verse-Bench Set 1, where Synchformer-C is the predicted synchronization confidence (higher is better), and Synchformer-O is the temporal offset in seconds (lower is better).
>
> > It would be more convincing to compare Ovi with a sequential baseline (e.g., Wan2.2 for T2V followed by MMAudio-L for V2A, or FishSpeech for TTS) to demonstrate the benefit of joint modeling.
>
> This is a great suggestion, and we have followed exactly the approach the reviewer described. Specifically, for videos intended to include human speech, we benchmark the audio-driven video generation approach: first, we use the SOTA text-to-speech model Fish Speech S1-Mini to generate monologues (choosing reference voices based on gender), and then utilize Wan2.2-S2V-14B to create video matching the audio. For videos which instead are intended to not contain speech, we adopt a video-driven audio generation strategy: we use Wan2.2-TI2V-5B to generate silent video and then employ MMAudio to create sound effects for the video. We refer to this method as AV-Pipe.
>
> Comparisons of Ovi and AV-Pipe can be found at https://anonymous-iclr-2026.github.io/sequential-comparison/, where we produce videos corresponding to 25 randomly selected evaluation prompts from Verse-Bench Set 1. We furthermore include this approach in our synchronization evaluations, which are provided below.
>
> |  | UniVerse-1 | JavisDiT | AV-Pipe | Ovi |
> |---|---:|---:|---:|---:|
> | Synchformer-C ↑ | 0.2732 | 0.3750 | **0.5024** | 0.4922 |
> | Synchformer-O ↓ | 0.86 | 0.59 | **0.48** | **0.48** |

---

> ### Comment · Reviewer_XP5Y · 2025-11-26
>
> Thank you for the detailed explanation and the additional experiment.
> While I acknowledge the differences between Ovi and prior work, my concerns have not yet been adequately addressed.
> My primary concern remains whether the reported performance improvements stem from the proposed modeling strategy or simply from upgrading the base model from OpenSora to Wan2.2.
> If the significance of Ovi primarily derives from Wan2.2's improved capability, then the contribution is largely engineering-driven rather than scientifically novel.
>
> For Part 1:
> I appreciate the authors' efforts to clarify the differences between Ovi and prior work.
> However, differences alone do not necessarily establish novelty.
>
> Regarding the conditioning mechanism, the authors did not provide a direct comparison between JavisDiT and Ovi. To support the claim that the proposed conditioning is beneficial, it would be more appropriate to measure the performance gains using the proposed method while keeping the base model (and ideally the dataset) fixed. Otherwise, the actual advantage and generalizability of the proposed design remain unclear.
>
> Regarding the differences between the proposed RoPE and MMAudio's, they stem from differences in input dimensionality rather than a fundamentally different algorithmic design.
> If I understand correctly, the proposed RoPE is applied along the temporal axes ($T$ and $T'$) of the video and audio representations, where video tokens have shape $(C,T,H,W)$ and audio tokens have shape $(C′,T′)$ with a temporal scaling factor.
> In MMAudio, the video tokens have shape $(C,T)$ (with no spatial dimensions), and RoPE is applied at $T$ and $T′$ using the same temporal scaling.
> Simply duplicating the RoPE across spatial dimensions seems trivial and, on its own, does not clearly establish novelty.
>
> For Part 2:
> Thank you for conducting additional experiments on audiovisual alignment.
> However, these results reinforce the same concern: the sequential generation pipelines (T2A + A2V or T2V + V2A) outperform Ovi, suggesting that the performance gains over UniVerse-1 and JavisDiT may largely come from the upgraded base model rather than the proposed joint modeling strategy.
> I also reviewed the updated user study and demo, but I was unable to identify a substantial improvement.
>
> Overall, these results do not justify the increased training cost or demonstrate a clear advantage of training a joint audio–video generation model with the proposed design over simply combining existing pre-trained video, audio, and speech generation models.

---

> > ### Author Response · Authors · 2025-12-01
> >
> > We thank you for the detailed comments.
> > 1. Ovi’s core contributions do not come from the video backbone. The key behaviors introduced by Ovi: intelligible speech generation, phoneme-aligned lipsync, music instrument and event-accurate sound effects are not present in Wan2.2. These behaviors arise from:
> > > - Our 5B audio diffusion transformer trained from scratch,
> > > - Our 1B bidirectional blockwise fusion trained from scratch, and
> > > - scaled cross-modal RoPE that enforces temporal structure without any prior estimators
> >
> > These design elements, not the video tower, drive Ovi’s emergent synchronization and speech–video coupling.
> >
> > 2.  Our qualitative side-by-side comparisons with links below using verse-bench prompt set clearly highlights Ovi’s strong audio, speech and music generation quality, as well as cross-modal synchronization capabilities, that cannot be attributed to the video backbone. We encourage the reviewer to take a closer look for the side by side comparison.
> > > - Ovi vs prior open models: https://anonymous-iclr-2026.github.io/ovi/
> > > - Ovi vs sequential pipelines: https://anonymous-iclr-2026.github.io/sequential-comparison/
> >
> > 3. Regarding the concern related to the RoPE between proposed method and MMAudio’s, our goal is not to propose a new positional encoding algorithm, but to show that consistent temporal encoding across heterogeneous audio, video tokens enables alignment to emerge naturally, without external priors or sync networks. We have demonstrated that our simplified, effective design approach with such RoPE design and cross-modal attentions can achieve superior synchronization and temporal alignment between audio and video modalities without training any prior estimators.

---

### Official Review · Reviewer_cBPT · 2025-11-01

**Soundness:** 3
**Presentation:** 3
**Contribution:** 3
**Rating:** 6
**Confidence:** 4

**Summary:**

This paper introduces OVI, a method for generating audio-visual content through a single feedforward pass, which avoids the need for post-hoc processing or a multi-step pipeline. The methodology involves two main stages. First, an audio tower is trained from scratch on a large-scale audio-visual (AV) dataset to ensure high-quality audio generation. Second, a twin-tower architecture, which utilizes blockwise bidirectional fusion and scaled Rotary Position Embeddings (RoPE), is trained to achieve strong synchronization and cross-modal alignment.

**Strengths:**

1. The paper is well-structured, and the methodology is presented with clarity, making it straightforward to understand.

2. The paper's contributions encompass multiple aspects of the problem, including data, architectural design, and training strategies.

**Weaknesses:**

1. The experimental evaluation appears to be missing key comparisons. The authors highlight cross-modal alignment as a primary strength of OVI; however, the empirical evaluation is predominantly focused on audio quality. The assessment of audio-video synchronization is not sufficiently demonstrated.

**Questions:**

1. To better substantiate the claims regarding audio-video synchronization, we suggest the authors benchmark OVI against relevant video-to-audio models (e.g., Diff-Foley [1], FoleyCrafter [2]). This would provide a more direct and comprehensive evaluation of the model's synchronization capabilities in comparison to other state-of-the-art approaches.

2. The large-scale video-audio dataset is a significant contribution and is crucial for enabling the reproducibility of the reported results. Could the authors clarify whether they plan to release the dataset or the associated data processing pipeline?

---

> ### Author Response · Authors · 2025-11-23
> **Rebuttal**
>
> We thank the reviewer for highlighting the clarity and breadth of our contributions and for their constructive feedback. We have revised our paper to address their critiques and hope the following clarifications will address the reviewer’s concerns.
>
> **1. Limited evaluation of audio–video synchronization**
>
> > - The authors highlight cross-modal alignment as a primary strength of Ovi; however, the empirical evaluation is predominantly focused on audio quality. The assessment of audio-video synchronization is not sufficiently demonstrated.
> > - To better substantiate the claims regarding audio-video synchronization, we suggest the authors benchmark Ovi against relevant video-to-audio models.
>
> We appreciate the reviewer’s helpful suggestion and would like to share the work we have done to improve our synchronization evaluations. While the original submission included human Pairwise Win Rate (PWR) evaluation against JavisDiT and UniVerse-1, we agree that it lacked (1) exact descriptions of the questions to participants; (2) objective synchronization metrics for joint models and pipelined models (e.g., video-driven audio generation); and (3) a visual demonstration of the superiority of Ovi’s synchronization abilities.
>
> To address (1), our revised paper includes, verbatim, the questions posed to the participants in the PWR evaluations, covering audio quality, video quality, and AV sync (Sec. 5.3). Our PWR evaluations are also expanded to include videos which are generated via sequential pipelines, which are discussed below.
>
> To address (2), we obtain new objective sync metrics for Ovi, JavisDiT, UniVerse-1, and “pipelined” videos by performing evaluations on Verse-Bench Set 1 (encompassing both speech and non-speech scenarios) generated by each model. This “pipelining” strategy, which we refer to as AV-Pipe, tests the efficacy of sequential methods such as video-driven audio generation in accordance with the reviewer’s suggestion. Specifically, for videos intended to include human speech, we benchmark the audio-driven video generation approach: first, we use the SOTA text-to-speech model Fish Speech S1-Mini to generate monologues (choosing reference voices based on gender), and then utilize Wan2.2-S2V-14B to create video matching the audio. For videos which instead are intended to not contain speech, we adopt a video-driven audio generation strategy: we use Wan2.2-TI2V-5B to generate silent video and then employ MMAudio to create sound effects for the video. Objective synchronization metrics for all four approaches are given below.
>
> |  | UniVerse-1 | JavisDiT | AV-Pipe | Ovi |
> |---|---:|---:|---:|---:|
> | Synchformer-C ↑ | 0.2732 | 0.3750 | **0.5024** | 0.4922 |
> | Synchformer-O ↓ | 0.86 | 0.59 | **0.48** | **0.48** |
>
> The table showcases Synchformer results on Verse-Bench Set 1, where Synchformer-C is the predicted synchronization confidence (higher is better), and Synchformer-O is the temporal offset in seconds (lower is better). As shown above, Ovi obtains nearly identical synchronization as AV-Pipe. However, the actual quality of the video and audio generated is of a much higher caliber, as we show in our paper in Fig. 4 of PWR results, as well as in our visual results described next.
>
> To address (3), we create two anonymous websites: the first, https://anonymous-iclr-2026.github.io/ovi/, presents matching videos generated by Ovi, JavisDiT, and UniVerse-1, and these 25 samples are uniformly randomly selected to fit upload constraints and ease reviewer inspection, and are included as ZIPs in the supplementary materials as well; full 205-prompt results are available upon request; the second, https://anonymous-iclr-2026.github.io/sequential-comparison/, compares Ovi and sequential pipelines (AV-Pipe).
>
> **2. Data Processing Pipeline and Dataset Release**
>
> > Could the authors clarify whether they plan to release the dataset or the associated data processing pipeline?
>
> We thank the reviewer for their question. We will release model checkpoints and inference code, and plan to further release our data processing and filtering pipeline, as well as training scripts. Portions of our dataset may be made available subject to licensing review. Our firm goal is to provide the most complete open-source AV generation stack to date.

---

### Author Response · Authors · 2025-12-01
**General Response**

Dear Area Chair and Reviewers:

Thank you for your engagement throughout the discussion phase! While we were unable to receive further feedback from the reviewers, we have taken strong steps to ensure that their concerns are thoroughly addressed. As the discussion phase comes to an end, we would like to take this opportunity to summarize the utility of our method and the efforts we have made during the review period.

As we mention in the paper, Ovi is the first open-source system capable of jointly generating expressive speech, sound effects, and video with natural lipsync. While most open-source video generation models assume one modality is given or claim joint video-audio generation despite weak synchronization, uneven A/V quality, and high complexity, Ovi achieves superior results without external sync modules or post-hoc alignment tricks, showing that strong alignment can emerge purely from large-scale end-to-end training. Compared to prior open models, Ovi is the only system that shows real audio–video synchronization. Joint models such as JavisDiT and UniVerse can generate both modalities, but their audio is mostly generic ambience and fails on sync-critical prompts like intelligible speech or action-specific sound effects and neither aligns temporally or semantically with the video.

We are pleased to receive positive remarks from reviewers, such as:
- [XP5Y] “The proposed model achieves state-of-the-art quality compared to existing open-source joint audio-video generation models.”
- [cBPT] “The paper's contributions encompass multiple aspects of the problem, including data, architectural design, and training strategies.”
- [pr5F] “This approach successfully enables lip-synchronized speech generation.”

In response to reviewer feedback, we have made the following additions to our paper:
- Expanded and clarified PWR evaluation (Sec. 5.3). We now report the full Verse-Bench Set 1 setup (205 prompts across Ovi/JavisDiT/UniVerse-1/AV-Pipe), a 50-participant blind study with 100 prompt-matched pairs each, and include the survey questions verbatim.
- Added a strong sequential baseline (“AV-Pipe”). We implemented and evaluated a two-track pipeline baseline:
  - Speech prompts: Fish Speech S1-Mini TTS → Wan2.2-S2V-14B audio-driven video.
  - Non-speech prompts: Wan2.2-TI2V-5B silent video → MMAudio SFX generation.
- Introduced objective AV synchronization metrics (Sec. 5.4). We report SynchFormer confidence and offset on Verse-Bench Set 1 for Ovi, JavisDiT, UniVerse-1, and AV-Pipe.
- Released anonymous qualitative comparison sites
  - Joint-model comparison: https://anonymous-iclr-2026.github.io/ovi/
  - Sequential vs. joint comparison: https://anonymous-iclr-2026.github.io/sequential-comparison/

We encourage the community to view our qualitative comparison sites to directly observe Ovi’s advantages and its contributions to open-source AV generation.

Once again, we deeply appreciate the time and expertise you have shared with us and are happy to provide any further clarifications on Ovi!

---

### Meta-Review · Area_Chair_7XEW · 2026-01-04

**Summary:**

This paper introduces an audio–visual video generation pipeline with synchronized audio and video, dubbed OVI. The key technical idea is to employ synchronized audio and video diffusion transformer towers with T5-based text conditioning, where cross-modal blocks densely synchronize features across the two modalities. The model is trained on large-scale audio–visual corpora using a flow-matching objective. Experiments are presented on text-to-video, audio-to-video, and text-to-audio tasks, demonstrating compelling qualitative quality and strong numerical results, including human subjective evaluations. Qualitative videos provided in the supplementary material further demonstrate convincing quality and synchronization between the modalities.

**AC Comments:**

The paper proposes a large-scale dataset and a simplified audio–visual video generation architecture compared to JavisDiT, with strong qualitative results as shown in the supplementary material. However, reviewers identify several shortcomings, including: i) the lack of direct numerical comparisons with existing audio–visual video generation models; ii) the similarity of the proposed architecture to JavisDiT, without sufficient ablation studies to justify the design changes and to clarify similarities to aspects of MMAudio position embeddings; and iii) limited ablation studies on joint audio–visual quality.

The AC independently reviewed the paper and agrees with the reviewers that, while the work represents an impressive standalone effort in video generation, it is important to better contextualize the contributions relative to prior work. As such, the paper requires stronger experimental validation and more thorough ablation studies, and is not ready for acceptance in its current form.

**Reviewer Concerns:**

*Reviewer cBPT* raises an important concern that the quantitative studies in the paper are limited to audio generation quality and lack analysis of audio–visual synchronization, and recommends comparisons to Diff-Foley and FoleyCrafter. The reviewer also seeks a clear commitment from the authors regarding the public release of the audio–visual dataset.

*Reviewer XP5Y* argues that the presented pipeline lacks novelty due to significant resemblance to the JavisDiT architecture, and seeks clarification on the claimed limitations of JavisDiT—specifically, the need for a learned prior for audio–visual synchronization. The reviewer also questions the similarity of the RoPE scheme to MMAudio and the lack of evidence supporting the benefits of the proposed prompt conditioning. In addition, the reviewer requests more details on the subjective evaluation, notes the limitation of objective evaluations primarily to audio quality, and calls for clearer explanations of the reported numerical analyses.

*Reviewer pr5F* points out that the supplementary results lack comparisons to other methods, that the subjective evaluation is missing critical details, and that the objective evaluation omits important metrics and comparisons to several prior approaches.

**Reviewer Scores:**

**Reviewer cBPT:** The authors provide clarification on the human evaluation of video quality and audio–visual synchronization. Specifically, they include the exact questions posed to human evaluators and present additional comparisons on audio–visual synchronization using the Synchformer model, showing comparable performance to AV-pipe while achieving higher video generation quality. The authors also confirm the public release of data processing scripts and portions of the training set, as permitted by the license. However, the paper does not provide comparisons to FoleyCrafter or Diff-Foley.

[*AC’s take on the response*] The authors’ response appears to directly address some of the concerns raised by the reviewer; however, several requested comparisons remain unaddressed.

**Reviewer XP5Y:** The authors provide:

* a detailed account of the limitations of JavisDiT in learning synchronization priors via contrastive learning, although no numerical comparisons are presented. The authors also state that the Ovi architecture enables training on much larger-scale data (37M waveforms, 2.5M sound effects, and 12M videos, compared to fewer than 1M samples in JavisDiT);

* clarification on the proposed prompt-conditioning scheme by pointing to the ablation study in Table 5, where Ovi with CLAP encoders is compared to T5, although no direct comparison to JavisDiT is provided to substantiate the claims;

* an explanation of the differences from MMAudio’s RoPE embeddings, noting that MMAudio uses per-frame positional encodings, whereas the scaled-RoPE in Ovi enables alignment between compressed spatio-temporal volumes and 1D audio tokens, again without numerical validation; and

* additional details on the subjective study, more qualitative results, and Synchformer experiments evaluating audio–visual synchronization.

[*AC’s take on the response*] The authors provide a reasonable rebuttal to the reviewer’s concerns; however, claims regarding comparisons to closely related approaches are largely explanatory and not supported by empirical evidence. While the paper aims to simplify architectural choices from prior work, particularly JavisDiT, it does not convincingly demonstrate performance advantages through direct comparisons. As such, the reviewer is unlikely to revise the recommended score.

**Reviewer pr5F:** The authors present additional qualitative results, including comparisons to JavisDiT and UniVerse-1, provide further details of the subjective study, explain why the FVD metric was not used, present synchronization experiments using Synchformer, and point to ablation studies. However, as noted above, the paper still lacks direct quantitative comparisons to audio–visual video generation models.

[*AC’s take on the response*] The response does not appear to adequately address the reviewer’s concerns regarding the absence of quantitative comparisons.

---

### Decision · Program_Chairs · 2026-01-26

Reject